# Scanning Probe Spectroscopy of WS_2_/Graphene Van Der Waals Heterostructures

**DOI:** 10.3390/nano10122494

**Published:** 2020-12-11

**Authors:** Franco Dinelli, Filippo Fabbri, Stiven Forti, Camilla Coletti, Oleg V. Kolosov, Pasqualantonio Pingue

**Affiliations:** 1CNR, Istituto Nazionale di Ottica, via Moruzzi 1, 56124 Pisa, Italy; 2NEST, Scuola Normale Superiore, Piazza San Silvestro 12, 56127 Pisa, Italy; filippo.fabbri@nano.cnr.it (F.F.); pasqualantonio.pingue@sns.it (P.P.); 3NEST, Istituto di Nanoscienze-CNR, Piazza San Silvestro 12, 56127 Pisa, Italy; 4CNI@NEST, Istituto Italiano di Tecnologia, Piazza San Silvestro 12, 56127 Pisa, Italy; stiven.forti@iit.it (S.F.); camilla.coletti@iit.it (C.C.); 5Graphene Labs, Istituto Italiano di Tecnologia, Via Morego 30, 16163 Genova, Italy; 6Department of Physics, University of Lancaster, Bailrigg, Lancaster LA1 4YB, UK; o.kolosov@lancaster.ac.uk

**Keywords:** 2DM, tungsten disulfide, epitaxial graphene, SiC, PF-QNM, UFM, EFM

## Abstract

In this paper, we present a study of tungsten disulfide (WS_2_) two-dimensional (2D) crystals, grown on epitaxial Graphene. In particular, we have employed scanning electron microscopy (SEM) and µRaman spectroscopy combined with multifunctional scanning probe microscopy (SPM), operating in peak force–quantitative nano mechanical (PF-QNM), ultrasonic force microscopy (UFM) and electrostatic force microscopy (EFM) modes. This comparative approach provides a wealth of useful complementary information and allows one to cross-analyze on the nanoscale the morphological, mechanical, and electrostatic properties of the 2D heterostructures analyzed. Herein, we show that PF-QNM can accurately map surface properties, such as morphology and adhesion, and that UFM is exceptionally sensitive to a broader range of elastic properties, helping to uncover subsurface features located at the buried interfaces. All these data can be correlated with the local electrostatic properties obtained via EFM mapping of the surface potential, through the cantilever response at the first harmonic, and the dielectric permittivity, through the cantilever response at the second harmonic. In conclusion, we show that combining multi-parametric SPM with SEM and µRaman spectroscopy helps to identify single features of the WS_2_/Graphene/SiC heterostructures analyzed, demonstrating that this is a powerful tool-set for the investigation of 2D materials stacks, a building block for new advanced nano-devices.

## 1. Introduction

Building nano-devices, based on vertical or lateral heterojunctions made of two-dimensional materials (2DM), is gaining increasing interest for electronics and photonics applications [1,2,3]. The production of these van der Waals heterostructures requires either transferring or the direct growth of individual 2DM layers, one on another or one besides the other [4,5]. The procedures developed to obtain ever more complex stacks need to be carefully planned, controlled, and verified step by step. In particular, each step has to be characterized taking into account several points of view, starting from the morphology and, ultimately, ending with the local physical properties. For instance, in a recent paper, we have demonstrated that the investigation should be carried out over reasonably short or long timeframes, in order to minimize any possible change induced by the exposure to the normal environment or to characterize any possible degradation phenomenon due to the operational conditions [6].

A standard investigation of 2DM samples generally begins with the employment of optical and electronic microscopies, such as scanning electron microscopy (SEM), and of spectroscopy techniques, such as photoluminescence and µRaman [7,8]. However, once the formation of the desired structures has been confirmed, other instruments are needed in order to access the physical properties on the nanoscale. These properties range from simple morphology to elastic [9] and electrostatic ones [10]. In addition, a complete characterization should also include the study of subsurface features, such as the conditions of the buried interfaces, that can strongly affect the elastic and the electrical properties as well as the desired performances of these heterostructures [11].

Up to now, investigations of 2DM layered samples have been generally performed mainly focusing on one or two specific physical properties. In this work, we have decided to use a broad range of complementary scanning probe microscopy (SPM) modes, in combination with SEM and µRaman spectroscopy, with the aim of demonstrating that a comprehensive analysis can be achieved. In particular, we report the case study of a sample made of a tungsten disulphide (WS_2_) layer grown by chemical vapour deposition onto epitaxial Graphene on a silicon carbide (SiC) substrate [12].

## 2. Materials and Methods

Epitaxial graphene (epiGr) was obtained via solid-state thermal decomposition of atomically flat, nominally on-axis 6H–SiC(0001) (SiCrystal GmbH, Nürnberg, Germany), employing a resistively-heated cold-wall reactor (modified commercial setup BlackMagic, Aixtron Ltd., Cambridge, UK) [13]. Tungsten disulphide (WS_2_) films were then deposited directly on epiGr/SiC via a low-pressure chemical vapour deposition (CVD) process, using tungsten trioxide (WO_3_) (99.995%, Sigma Aldrich, St. Louis, MO, USA) and sulphur (S) (99.998%, Sigma Aldrich, St. Louis, MO, USA) as precursors [14]. The process was performed in a 2.5 inches horizontal hot-wall PTF furnace (Lenton, Hope Valley, UK). The furnace can be described as follows: a central hot-zone, where a crucible loaded with WO_3_ powder was lodged at 2 cm from the substrate; an inlet zone, where the Sulphur powder was positioned and heated by a resistive belt. The powder was heated at 180 °C and the S vapour reacted with the WO_3_ molecules directly on the substrate surface at a temperature of 900 °C and a pressure around 5 × 10^−2^ mbar, using Ar as a carrier gas. The samples were kept in opaque containers under a nitrogen atmosphere prior to any analysis, and in opaque containers in air during the SPM analysis.

The SPM analysis was performed using two different setups, independently calibrated with standard specimens. The first one is a commercial Icon system, equipped with a Nanoscope V controller (Bruker, Billerica, US). It was employed in the peak force–quantitative nano mechanics (PF-QNM) configuration [15]. This mode allows one to record simultaneously topography and mechanical properties (adhesion, Young’s modulus, deformation, dissipation) [16]. The second setup is a hybrid system, made of a commercial SMENA head (NT-MDT, Moscow, Russia), home-built electronics and a HF2LI digital lock-in amplifier (Zurich Instruments, Zurich, Switzerland) [11]. It was operated in ultrasonic force microscopy (UFM) [17,18], Kelvin probe force microscopy (KPFM) [19], and dielectric force microscopy (DFM) [20].

The cantilevers employed were: “scanasyst-air” model (Bruker, Billerica, MA, USA) for PF-QNM, CSG30 (NT-MDT, Moscow, Russia) for UFM, and PtSi-FM (Nanosensors, Neuchatel, Switzerland) for KPFM and DFM (first two resonant modes at 77 kHz and 489 kHz). The sample mechanical excitation in UFM was obtained via a piezoelectric ceramic actuator (PI Ceramic GmbH, Lederhose, Germany), driven by a 33-220A function generator (Agilent, Santa Clara, CA, USA). The output of the lock-in amplifier was used for the electrical excitation in KPFM and DFM. Finally, the image processing was performed using a free software, Gwyddion [21].

Scanning electron microscopy (SEM) is typically performed to initially characterize the morphology on a large scale. In our case, this analysis was carried out in a Merlin FEG-SEM microscope (Zeiss, Jena, Germany), with an accelerating voltage of 5 kV and an electron current of 12 pA, employing both standard “in lens” and Everhart-Thornley detectors, respectively for high angle (SE1) and low angle (SE2) type secondary electron detection, in order to increase the spatial resolution at the surface and to minimize the damage of the sample due to the electron beam irradiation.

Scanning µRaman spectroscopy was carried out with a InVia system (Renishaw, Wotton-under-Edge, UK), equipped with confocal microscope, a 532 nm excitation laser, and an 1800 line/mm grating (with a spectral resolution of 2 cm^−1^). All the measurements were performed with the following parameters: excitation laser power 1 mW, acquisition time for each spectrum 2 s, and pixel size of 1 µm × 1 µm.

## 3. Results

Two-dimensional material (2DM) samples after the growth process are routinely analyzed by means of scanning electron microscopy (SEM). In Figure 1, we show an example of such an analysis, carried out on a sample made of a tungsten disulphide (WS_2_) layer CVD-grown on epitaxial Graphene (epiGr/SiC). The WS_2_ crystals can be clearly individuated both in the SE2 (Figure 1a) and SE1 (Figure 1b) imaging modes [22], thanks to their characteristic triangular shape. The presence of the crystals can be revealed on a wide scale and their nucleation densities easily studied. Besides the WS_2_ triangular terraces, irregular stripes crossing the whole area are visible with a dark contrast in the SE2 image and a bright contrast in the SE1 one. Other features, lines, and dots located on the crystal edges also present a dark contrast in the SE2 image and a bright one in the SE1 image.

Besides SEM, a µRaman analysis is typically performed on 2DM samples for the evaluation of the number of layers. The two spectra, shown in Figure 1b, were taken in different areas of the sample. They present the standard Raman peaks of WS_2_: the superimposed 2LA(M) and E2g(Γ) peaks, at 351 cm^−1^, and the A1g(Γ) peak at 418.5 cm^−1^. In particular, it is possible to individuate single layers (red line) and bi-layers (green line). The µRaman map, in Figure 1d, evidences the presence of a continuous WS_2_ monolayer with bilayer terraces. This characterization is however limited by a low spatial resolution. Recent developments, namely tip enhanced Raman spectroscopy (TERS) and photoluminescence (TEPL), are very promising in order to overcome this limitation [23,24].

SEM imaging and µRaman mapping can be thus considered very useful for a large area complementary characterization of the sample. It is however not possible to obtain other relevant information at the sub-micrometric scale, such as the height of few-layer nanoscale terraces, the mechanical and electrostatic properties, the localization of extended defects and the origin of the SEM contrast in some specific areas. The main topic of this work deals with an experimental approach devised to improve the sample analysis at the sub-micrometric scale, given that a characterization only based on SEM and µRaman cannot be sufficiently accurate. In the following, we intend to prove that using some specific SPM techniques the topographical, elastic, and electrostatic properties of 2DM heterostructures can be better defined, allowing a proper physical characterization and, finally, an effective exploitation of their electronic and photonic properties.

In Figure 2, we first present some data obtained with peak force–quantitative nano mechanics (PF-QNM) on the same sample of Figure 1. In this mode, the tip is approached to and retracted from the surface in a number of points, typically equally spaced and selected over a square area [15]. The data collected allow one to simultaneously realize spatially resolved maps of morphology, adhesion, deformation, and dissipation properties [16]. In our case, the topography shows the characteristic triangular WS_2_ crystals on a stepped surface morphology, typical of the epiGr/SiC substrates. The height of the triangular terraces is 0.5 ± 0.2 nm, whereas the other visible steps, identified with the SiC substrate underneath, measure 0.7 ± 0.2 nm (Figure 2b). The crystal edges have a thickness from 2 to 4 nm, whereas the height of other bumps ranges from 10 to 30 nm.

The adhesion channel (Figure 2c) presents the crystal edges and the bumps as dark features. This behaviour suggests that they may have different adhesive properties with respect to the WS_2_ surface; not to exclude however some contributions of topography. As shown in Appendix A, a fresh sample does not present a thickening of the crystal borders, a phenomenon due to the oxidation process that occurs immediately after an exposure to the ambient environment, and initiated at local defects in the crystal edges [6]. The presence of irregular stripes, also noticed in the SEM image of Figure 1b, is confirmed. These features in SPM are depressed in topography and can be identified as trenches on the surface. Apart the dark contrast due to the oxidation at their edges, no other features or contrast can be observed at the bottom of these areas in the adhesion channel.

In Figure 3, we now present some data obtained by means of ultrasonic force microscopy (UFM), on a sample grown and aged in the same conditions as the one of Figure 1 and Figure 2. In this imaging mode, the tip is in contact with the surface at a given load and the sample is vertically ‘shaken’ at an ultrasonic frequency, while scanning the area under investigation [17,18]. The effective cantilever stiffness at a MHz frequency is significantly higher than the static value [25]. This characteristic is very useful for mapping the elastic properties of 2DMs grown on a solid substrate and the image colour code is the following: the darker the colour, the higher the local deformation.

The topography channel reveals a morphology with the characteristic triangular WS_2_ terraces (see also Appendix A that includes line profiles). Their height (point 1) is equal to 0.5 ± 0.2 nm. The elongated terraces are 0.7 ± 0.2 nm thick (point 3), and can be identified with SiC. Some topographical “bumps” are again visible in the corners of the triangular terraces and some thin decoration along their edges. The height of these bumps ranges from 10 to 30 nm. The irregular trenches, shown in Figure 1 and Figure 2, are still present. They have a depth of 0.8 ± 0.2 nm (point 2) and are decorated as the crystals by darker features in the elastic map. This oxidation process is a fingerprint of the WS_2_ presence and suggest, as also indicated by µRaman mapping, that the surface of the sample is totally covered with a WS_2_ continuous layer, on top of which the WS_2_ triangular islands form an additional layer.

Moreover, in the elastic map, a dark contrast can be clearly observed at the bottom of the trenches, compared to the rest of the surface. The elastic deformation is dependent on the total load, equal to the load applied via the cantilever plus the tip/surface adhesion. According to the adhesive map in Figure 2b, there is no noticeable difference between the trenches and the flat areas. This may indicate a true difference in the elastic behavior. The depth of the trenches indicates that epiGr is here exposed, not the buffer or the SiC substrate. Therefore, we believe that the trenches can be identified with cracks opening in the WS_2_ surface during the cooling phase after the growth process of WS_2_, thus exposing epiGr to the surface. As the depth of the trenches is not fully homogeneous, it may occur that also the epiGr has cracked in some places.

Finally, depending on the scanning conditions of the probe (i.e., applying larger loads and switching off the ultrasonic vibration), some wear of the surface can be observed. This wear process occurs at the edges but also within and around the triangular terraces. This behaviour may be explained with the detachment of portions of the oxidized edges of the WS_2_ crystals and/or with probe induced “sweeping” of residual WS_2_ precursors or oxide particles from the sample surface (see Appendix A).

The same sample of Figure 3 was also analyzed from an electrostatic point of view. In Electrostatic Force Microscopy (EFM), a conductive tip is biased with respect to the sample, which determines a force due to capacitive effects [26]. A bias with a DC and an AC component is applied as a driving element. In our experiments, we employed a “single pass” mode [27]: the resonance frequency is mechanically excited, and the amplitude is used to feedback the distance of the probe from the surface of the sample, as in tapping mode; the tip is simultaneously AC biased at a frequency equal to the first overtone of the cantilever or to the half of it. The detection may be carried out with via a lock-in amplifier at the first or second harmonic of the bias excitation frequency. In the first case, the signal is sensitive to the difference between the work functions of the tip and the sample, and hence related to the local Fermi level of the surface: this technique is generally known as Kelvin probe force microscopy (KPFM) [19]. KPFM can be operated in more sophisticated approaches, that can be used in other circumstances where one needs to carry out a more accurate and quantitative analysis [28]. In our case, the “single pass” mode is sufficiently sensitive in order to obtain a qualitative contrast to be cross-correlated with UFM.

In Figure 4a,b, corresponding to the topographic signal, the patterns of the oxidation are always visible at the edges of the WS_2_ crystals. The height the WS_2_ crystals results equal to 0.55 ± 0.2 nm (point 1), whereas the SiC steps are 0.65 ± 0.2 nm thick (point 3). In this area, the irregular trenches have a depth of 0.75 ± 0.2 nm at point 2 and of 0.25 ± 0.2 nm at point 2**′**. The image clearly shows that the depressed area at point 2**′** is rougher than at point 2.

In the corresponding first harmonic channels (Figure 4c,d), the WS_2_ crystals present a bright contrast with respect to the background. This contrast becomes very strong in correspondence of the trenches, which are however not homogeneous. At point 2 the more depressed regions are almost as bright as the rest, whereas at point 2**′** the less depressed regions are dark. In addition, we can observe that within each single WS_2_ crystal the contrast may finely change.

Finally, it can be stated that the oxidized regions are always darker with respect to WS_2_. If one applies an AC bias ≥5 V, the oxidized borders can be dramatically modified with the appearance of more bumps decorating the crystal edges, as shown in Appendix A.

When detection is carried out at the second harmonic of the bias frequency, the signal is sensitive to the local dielectric permittivity: this technique is generally known as dielectric force microscopy (DFM) [20]. In Figure 5, we show some data of the first and second harmonic channels, taken in the same region. First, we need to underline that the contrast is quite different and somehow unexpected. The bumps and the oxidized regions are better enhanced in DFM with respect to KPFM. In the second harmonic map, there is however no clear difference between the WS_2_ terraces and the background, with the exclusion of the second order terraces. This is not in agreement with the first harmonic channel that, as previously underlined, shows a clear and complex contrast among regions within the same or different terraces and the trenches. The DFM data might thus indicate that the whole area is covered with WS_2_. Nevertheless, the trenches do not show any noticeable contrast either.

## 4. Discussion

The following discussion of our results will be focused on three main aspects: on the one hand the morphology and height measurements, on the other the mechanical and electrostatic contrasts. All these parameters may dramatically change depending on the growing conditions, the degree of oxidation, and the degradation process of the samples.

First of all, SEM and SPM can both identify the presence of triangular terraces on a stepped substrate with a morphology made of elongated terraces. The triangular features can be attributed to WS_2_ monolayer crystals, as confirmed by the µRaman analysis (Figure 1), and the other elongated terraces to the SiC substrate underlying the epiGr layer. Whereas SEM can quickly explore wide areas with high lateral resolution, SPM can measure the vertical dimension and in addition the mechanical and electrostatic properties. The height of the WS_2_ islands can be quantified, by means of both the two SPM setups employed in this work, in 0.5 and 0.55 nm respectively, with an uncertainty of 0.2 nm. The second setup has been operated both in contact and tapping modes, yielding nearly the same values. It must be underlined that our setups have been independently calibrated and are located in different laboratories.

In the literature, the thickness measured of a WS_2_ sheet on a WS_2_ substrate is equal to 0.6 nm [8,9,10,11,12,13,14,15,16,17,18,19,20,21,22,23,24,25,26,27,28,29]. Therefore, our data are consistent with what previously reported, considering the error bar. The theoretical thickness of a WS_2_ sheet, i.e., the distance between the top and bottom planes made of S atoms, has been calculated in 0.32 nm [30]. Our measurements indicate that the distance between the bottom S plane of the WS_2_ top layer and the top S plane of the WS_2_ bottom layer can be thus estimated in 0.2 ± 0.2 nm.

It has been reported that the thickness may depend on the density of defects [31]. As the oxidation process determines an increase of the defect density, this can induce a reduction of the crystal average thickness. In particular, we have directly verified, using the SPM spatially-resolved scratching method (see Appendix A), that at the end of the oxidation process, i.e., after few weeks of exposure to the ambient environment, the features with a triangular shape are formed by thicken rims, whereas inside it is not possible to measure any step. This strongly suggests that there might be no WS_2_ left due to the oxidative degradation.

Secondly, SEM and SPM can all observe the thickening of the borders within few days of the ageing time. Their thickness can be evaluated with both SPM setups, giving the same range of values. The decoration of the edges, the trenches, and the bumps at the corners of the terraces have been also observed with both SEM and SPM techniques. In this case as well, the thickness can be only evaluated with SPM. The bumps can be considered fingerprints of the nucleation sites, whereas the trenches can be identified with cracks formed during the thermal stress that the sample undergoes during the deposition and the final cooling [6].

Finally, it must be underlined that the experimental evaluation of the thickness of a WS_2_ monolayer may depend not only on the SPM mode employed [32] but also on the nature of the substrate where the crystals have been transferred to or grown onto. This may explain the difference of thickness measured in the trenches where WS_2_ lies on epiGr/SiC. In Figure 3 and Figure 4, the trench thickness is evaluated in 0.8 and 0.75 nm respectively, with an uncertainty of 0.2 nm. This is also in agreement with the measurements of WS_2_ lying on different substrates reported in the literature [8,29].

With regard to the mechanical properties, PF-QNM is very sensitive to adhesion. On the contrary, UFM is very sensitive to the local indentation but this physical parameter can be affected by variations in both elasticity and adhesion. Therefore, the use of both techniques helps to better characterize the specimens and to have a reliable interpretation of the UFM contrast. For instance, the oxidized borders and bumps present a higher adhesion according to PF-QNM and they are also more compliant than the surrounding areas according to UFM. As reported elsewhere, a higher adhesion determines an increase of the compliance measured via UFM [25]. Therefore, the WS_2_ crystals have genuinely a higher elastic modulus than the oxidized borders, although the topography may also give a contribution.

The trenches, on the other hand, are more compliant than the WS_2_ crystals according to UFM. In this case, the contrast is not influenced by either topography or the local adhesion, as determined from the PF-QNM analysis. Thus, the regions of the WS_2_ crystals should have an elastic modulus higher than the regions inside the trenches. In the literature, graphene is reported to have a higher Young’s modulus than WS_2_ [33]. Those measurements have been performed in a free-standing configuration, where 2D flakes are suspended over holes. Our configuration is completely different as graphene and WS_2_ have been directly grown on a substrate. This might explain the discrepancy that we find: we actually measure the elasticity of a layered structure, that has undergone a strong thermal treatment.

In addition, UFM allows one to be subsurface sensitive, at least in the volume affected by the elastic field induced by the load applied through the tip. In such a way, UFM can detect the “quality” of the buried interfaces. For instance, this can apply in the regions where a WS_2_ triangular terrace lies directly on epiGr/SiC or on another WS_2_ terrace. In the case herein reported, the buried interfaces can be generally described as well adherent. However, in some small areas a darker contrast in the elastic map may indicate a reduced adhesion, such as the case of the place indicated with “α” in Figure 3d.

With regard to the electrostatic properties, KPFM measures the local Fermi level, while DFM measures the dielectric constant. From our data, their contrasts are peculiar and not easy to be correlated to the morphology. For instance, we found that the Fermi level can change within a single WS_2_ crystal and can be affected by the presence of the underlying terraces. It can also change within the cracks where, according to UFM and the height calculated in Figure 3, WS_2_ may not be present. One needs however to exclude the case of point 2**′** shown in Figure 4, where the height is equal to only 0.25 nm. We suggest that, in this region, some WS_2_ might still be present.

These variations in the KPFM contrast may be thus attributed to discontinuities at the buried interfaces, as some elastic non-homogeneities have been also observed with UFM. Finally, we need to stress that the dielectric constant seems to only evidence the areas where oxidation occurs or where some loose residual material may have remained after the deposition process.

## 5. Conclusions

Herein, we have presented a study of tungsten disulphide films grown on epitaxial Graphene with a variety of scanning probe microscopy techniques combined to standard scanning electron microscopy and µRaman spectroscopy. In particular, combining data obtained in contact and tapping modes has allowed us to accurately characterize the morphology. Peak force–quantitative nano mechanics has permitted the detection of surface properties, such as adhesion, whereas ultrasonic force microscopy has permitted to map the elastic properties of the sample, including subsurface features. Fine details, within single and multilayer terraces, can be reasonably attributed to the conditions of the buried interfaces. Finally, electrostatic force microscopy, working either at the first or the second harmonics, has allowed us to characterize the electrostatic properties of these heterojunctions. The proposed approach of combining all these techniques represents a powerful tool-set for those who aim at building and characterizing in-situ layered structures to be employed as active regions in advanced nano-devices. We envision that it could be applied in the future to finely tune the growth process of the 2D heterostructures in function of the final application.

## Figures and Tables

**Figure 1 nanomaterials-10-02494-f001:**
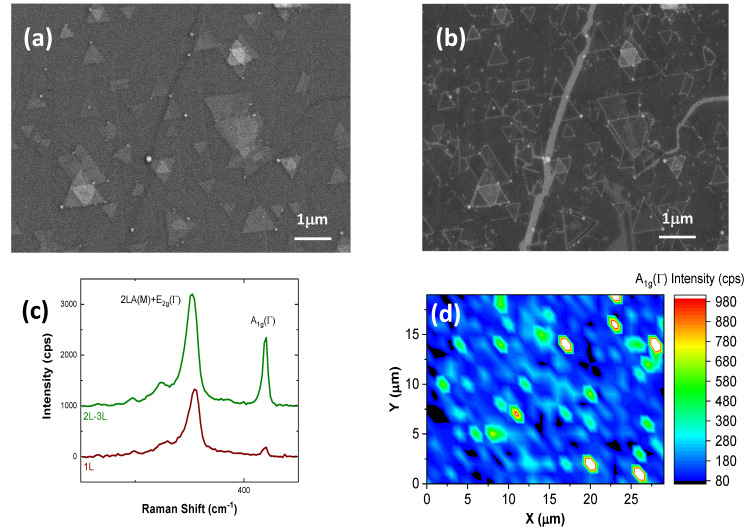
SEM (5 keV) image of a sample made of a WS_2_ layer CVD-grown on epiGr/SiC, taken after 2 days of exposure to the ambient environment. (**a**) Image obtained from the SE2 detector; (**b**) image obtained from the SE1 detector. Bright dots and lines can be observed on the edges of the triangular WS_2_ crystals. Long and irregular stripes can also be noticed in both images. (**c**) Characteristic µRaman spectra taken in different areas of the sample and (**d**) µRaman map of the A1g(Γ) mode intensity: the dark spots are due to the absence of WS_2_.

**Figure 2 nanomaterials-10-02494-f002:**
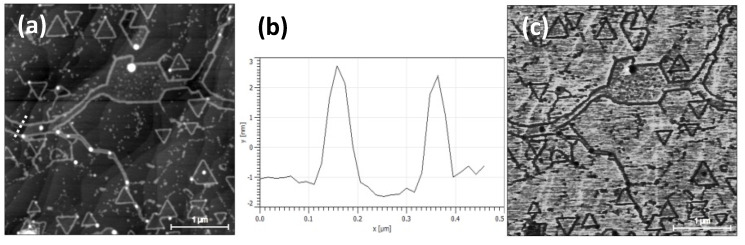
PF-QNM data obtained on the same sample of Figure 1. Specifically, (**a**) morphology channel; (**b**) height profile of the dotted line drawn in (**a**); (**c**) adhesion channel. Height values: triangular terraces, 0.5 ± 0.2 nm; crystal edges, from 2 to 4 nm; bumps, from 10 to 30 nm.

**Figure 3 nanomaterials-10-02494-f003:**
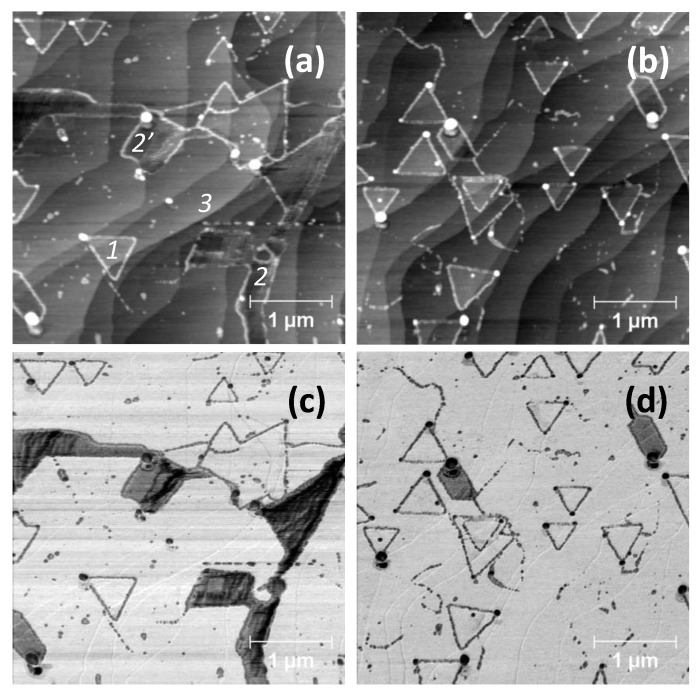
UFM data obtained on two different areas of a sample grown and aged in the same conditions of the samples in Figure 1 and Figure 2: (**a**,**b**) morphology and (**c**,**d**) the corresponding elastic channels. Height values: triangular terraces, 0.5 ± 0.2 nm (point 1); crystal edges, from 2 to 4 nm; bumps, from 10 to 30 nm.

**Figure 4 nanomaterials-10-02494-f004:**
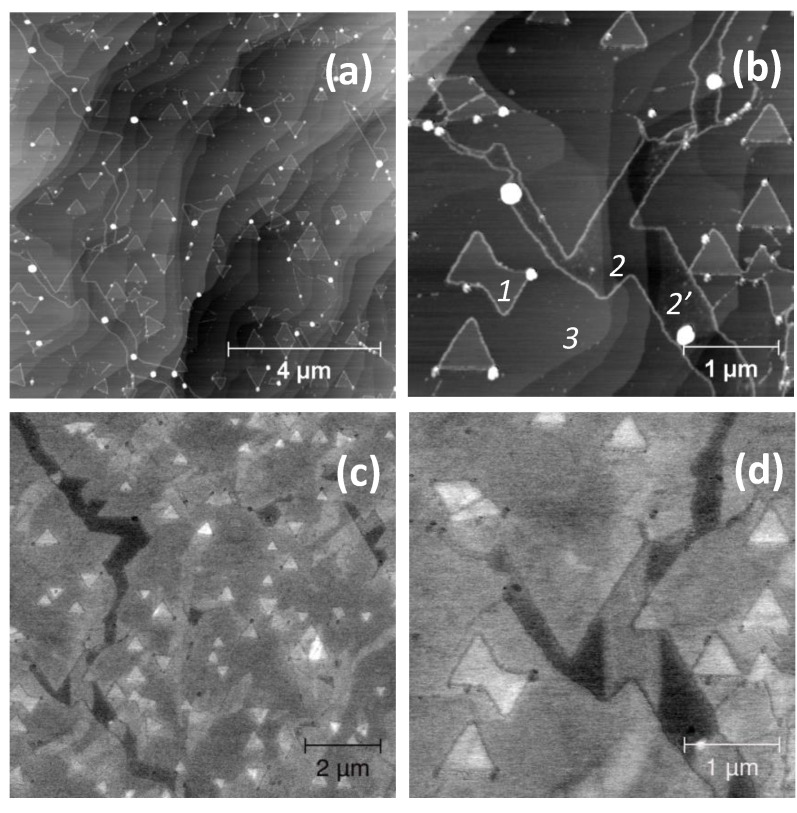
KPFM data obtained on the same sample of Figure 3: (**a**,**b**) topography at two different magnification and (**c**,**d**) the corresponding first harmonic channel maps, obtained applying 1 V to the tip, at a frequency of 489 kHz.

**Figure 5 nanomaterials-10-02494-f005:**
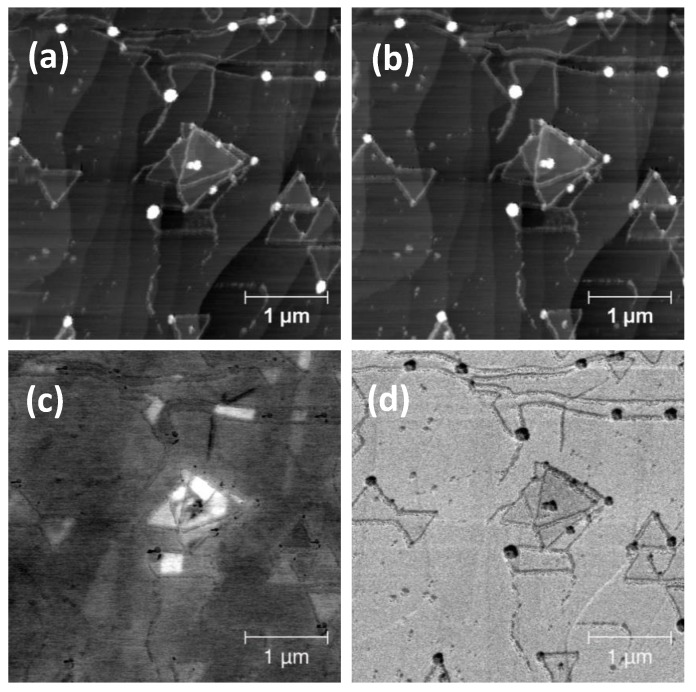
KPFM and DFM data obtained on the same sample of Figure 4, in the same region: (**a**,**b**) morphology and the corresponding (**c**) first harmonic channel (KPFM) and (**d**) second harmonic channel (DFM). The electrostatic maps have been obtained applying 2 V to the tip, at a frequency of 489 kHz (**c**) and 244.5 kHz (**d**).

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
