# Peer review of "Scanning Probe Spectroscopy of WS2/Graphene Van Der Waals Heterostructures"

_nanomaterials, 2020, doi:10.3390/nano10122494_

Round 1
Reviewer 1 Report
Dinelli and coworkers report different physical magnitudes of a WS2 sample grown by CVD on an epiGr/SiC. They used SEM and Raman for initial catheterization, but they obtain the most important results by using different AFM modes. I found the manuscript exciting and appropriate for the readership of Nanomaterials journal. There are, however, several remarks and suggestions that I think must be considered in detail before publication:
- The authors use Peak Force microscopy to measure both topography and adhesion. Peak force is a commercial name from a company, but the physics underlying this scanning mode can be traced back, for instance, to Jumping Mode scanning force microscopy by P. J de Pablo et al. https://doi.org/10.1063/1.122751.
- Quoting the authors (lines 164) "This behavior proves that they are different materials." I would not be so enthusiastic about adhesion maps. AFM magnitude maps, such as adhesion or many others, present a significant topographical contribution. This contribution is usually important right at the edges, where the feedback error tends to be high. Hence, just from the adhesion maps, this conclusion seems a bit speculative.
- Figure 3 portrays several AFM images of the sample mentioned above. Now, in addition to topography, the authors measured magnitude maps of mechanical properties. By comparing morphology images and elastic maps, they conclude that most of the sample is covered by WS2. Besides, there are uncovered areas (the trenches) exposing epiGr and not the SiC substrate. Based on the elastic maps, they also conclude that the trenches present a lower young modulus than the WS2 I do not understand this claim. The elastic moduli of graphene (~340 N/m) is twice the young modulus of WS2 (~170). Please elaborate further on this point. In fact, I think it will be very illustrative and, in my opinion, highly important to support the claims of the manuscript to prepare a sample with a microexfoliated flake of WS2 adsorbed on a graphite surface. I wonder if you would obtain qualitative and quantitave agreement with the data reported in the manuscript.
- The electrostatic images shown in figure 5 are also puzzling to me. I do not have either a clear understanding of the contrast. In my opinion, the experiment suggested above using a microexfoliated flake can also sed some light on this issue.
- A word about data presentation: I would include 2D-profiles (even in the supplementary information if you want) to clarify many of the manuscript's issues regarding to heights and measurements in the images.
Author Response
Dinelli and coworkers report different physical magnitudes of a WS sample grown by CVD on an epiGr/SiC. They used SEM and Raman for initial catheterization, but they obtain the most important results by using different AFM modes. I found the manuscript exciting and appropriate for the readership of Nanomaterials journal. There are, however, several remarks and suggestions that I think must be considered in detail before publication:
- The authors use Peak Force microscopy to measure both topography and adhesion. Peak force is a commercial name from a company, but the physics underlying this scanning mode can be traced back, for instance, to Jumping Mode scanning force microscopy by P. J de Pablo et al. https://doi.org/10.1063/1.122751.
We apologize for not having cited the original paper along with the reference relative to the commercial system we have employed in this work. De Pablo et al. are now cited as Ref. [16].
- Quoting the authors (lines 164) "This behavior proves that they are different materials." I would not be so enthusiastic about adhesion maps. AFM magnitude maps, such as adhesion or many others, present a significant topographical contribution. This contribution is usually important right at the edges, where the feedback error tends to be high. Hence, just from the adhesion maps, this conclusion seems a bit speculative.
The reviewer is correct in stating that the topography may affect the evaluation of other properties such as adhesion, as the contact area between the tip and the surface may differ. We have changed the text (line 159): “The adhesion channel (Figure 2c) presents the crystal edges and the bumps as dark features. This behaviour suggests that they may have different adhesive properties with respect to the WS2 surface; not to exclude however some contributions of topography. As shown in Figure S1, a fresh sample does not present a thickening of the crystal borders, a phenomenon due to the oxidation process that occurs immediately after an exposure to the ambient environment, and initiated at local defects in the crystal edges [6].”
- Figure 3 portrays several AFM images of the sample mentioned above. Now, in addition to topography, the authors measured magnitude maps of mechanical properties. By comparing morphology images and elastic maps, they conclude that most of the sample is covered by WS .
This conclusion is not only based on the elastic maps but on the basis of the mRaman data and the formation of the oxidized rims, phenomenon that occurs only at the WS2 crystal edges. We have added a short sentence (line 182): ”as also indicated by mRaman mapping”
- Besides, there are uncovered areas (the trenches) exposing epiGr and not the SiC substrate. Based on the elastic maps, they also conclude that the trenches present a lower young modulus than the WS I do not understand this claim. The elastic moduli of graphene (~340 N/m) is twice the young modulus of WS (~170). Please elaborate further on this point.
UFM can map the elastic deformation of this relatively stiff materials, otherwise not feasible with standard SPM modes. The elastic deformation is dependent on the total load, equal to the load applied via the cantilever plus the tip/surface adhesion. The Peak Force adhesive map in Figure 2b shows that there is no noticeable difference in adhesion between flat areas in the trenches and in the WS2 terraces. Thus, we may conclude that the regions inside the trenches present a higher a deformation and thus a lower elastic modulus. We agree that graphene is reported to have a higher elastic modulus than WS2. However those data refer to a different sample configuration, i.e. freestanding flakes (Liu, J. Mater. Res., Vol. 31, 2016). In our case, we have as substrate a CVD-grown graphene on SiC substrate undergoing a strong thermal treatment. This might explain our results.
We have modified the text (line 186): “The elastic deformation is dependent on the total load, equal to the load applied via the cantilever plus the tip/surface adhesion. According to the adhesive map in Figure 2b, there is no noticeable difference between the trenches and the flat areas. This may indicate a true difference in the elastic behavior. The depth of the trenches indicates that epiGr is here exposed, not the buffer or the SiC substrate. Therefore, we believe that the trenches can be identified with cracks opening in the WS2 surface during the cooling phase after the growth process of WS2, thus exposing epiGr to the surface. As the depth of the trenches is not fully homogeneous, it may occur that also the epiGr has cracked in some places.”
We have also modified the text (line 335): “In the literature, graphene is reported to have a higher Young’s modulus than WS2 [33]. Those measurements have been performed in a free-standing configuration, where 2D flakes are suspended over holes. Our configuration is completely different as graphene and WS2 have been directly grown on a substrate. This might explain the discrepancy that we find: we actually measure the elasticity of a layered structure, that has undergone a strong thermal treatment.”
- In fact, I think it will be very illustrative and, in my opinion, highly important to support the claims of the manuscript to prepare a sample with a microexfoliated flake of WS adsorbed on a graphite surface. I wonder if you would obtain qualitative and quantitave agreement with the data reported in the manuscript.
The reviewer suggests a very interesting experiment, which we have also planned at some stage of this work. We agree that we would gain more insight if we could investigate such a configuration as a reference sample, but we have not been able to realize it yet as it is not straightforward to make it. However, the aim of our manuscript is to show that a multi-technique approach can be useful for the characterization of this class of heterostructures. Nevertheless, as discussed at point #4, we can argue that an exfoliated configuration would be quite different in terms of the mechanical properties, in particular due to the nature of the interfaces. In our experience, exfoliating and transferring can produce a very different and not homogeneous interface between the different layers compared to an in-vacuum direct CVD growth.
- The electrostatic images shown in figure 5 are also puzzling to me. I do not have either a clear understanding of the contrast. In my opinion, the experiment suggested above using a microexfoliated flake can also shed some light on this issue.
As discussed at point #5, the exfoliated configuration might be quite different from the CVD grown stack also in terms of the electrostatic properties. This consideration makes even more interesting the idea of studying the exfoliated configuration suggested by the reviewer. It is a goal that can be tackled and achieved with the approach herein proposed.
- A word about data presentation: I would include 2D-profiles (even in the supplementary information if you want) to clarify many of the manuscript's issues regarding to heights and measurements in the images.
This is absolutely needed. We have added 2D line profiles to Figure 2 and S1.
Reviewer 2 Report
This is a good paper about the characterisation of WS2 triangles on the surface of epitaxial graphene. I recommend the following corrections for this paper;
- On page 3 line 125, the authors refer to the Raman map in figure 1c when this should be 1d.
- In Figure 2, could the authors add a line scan for the topography image in figure 2(a) so that it is obvious that the long lines running through the image are in fact trenches.
- On page 6 line 210, the authors use a bias on the tip for KPFM measurements, could the authors state what bias was used to obtain figure 4 and which frequency.
- Have the authors done any EDX analysis with the SEM to find out what the elemental composition of the oxidised regions are?
- The images in the paper do not have any gradient scales on the side to indicate e.g. height in the topography case. Could the authors add the colour scale on the side of the images so that values can be put to images.
Author Response
This is a good paper about the characterisation of WS2 triangles on the surface of epitaxial graphene. I recommend the following corrections for this paper:
- On page 3 line 125, the authors refer to the Raman map in figure 1c when this should be 1d.
We have amended the text.
- In Figure 2, could the authors add a line scan for the topography image in figure 2(a) so that it is obvious that the long lines running through the image are in fact trenches.
We have added a 2D line profile to Figure 2.
- On page 6 line 210, the authors use a bias on the tip for KPFM measurements, could the authors state what bias was used to obtain figure 4 and which frequency.
We have added the requested information to the captions of Figures 4 and 5.
- Have the authors done any EDX analysis with the SEM to find out what the elemental composition of the oxidised regions are?
In a previous work, we have reported a SEM-EDX analysis on these samples (see Ref. 6). However, it is not trivial to find a good trade-off for the accelerating voltage in order to have both surface sensitivity and a high signal-to-noise ratio. Actually, the accelerating voltage determines both the electron penetration depth and the element X-ray emissions. A higher accelerating voltage gives a better single-to-noise ratio, however the analysis is less surface sensitive. In our previous work, we have been able to detect the presence of oxygen after a period of 60 days, using an accelerating voltage of 7.5 keV. However, the tungsten related X-rays emission could not be excited. Thus, it has not been possible to accurately evaluate the stoichiometry of the oxidized areas.
- The images in the paper do not have any gradient scales on the side to indicate e.g. height in the topography case. Could the authors add the color scale on the side of the images so that values can be put to images.
We believe that the ‘z’ color scales are not accurate and the reader cannot gain much information from them. We prefer to report the height values of the relevant features, steps and bumps or rims. We have thus reported this information also in the captions of Figures 2 and 3. In addition, we have inserted 2D line profiles in Figures 2 and S1, which better serve to the purpose.
Reviewer 3 Report
The stated goal of demonstrating a comprehensive analysis was achieved for samples of WS2 on epi Graphene (epiGr) on SiC substrate, using an impressive set of techniques: two AFMs enabling PF-QNM, UFM, KPFM, DFM plus SEM, microRaman and PL.
The sample preparation is well described, including the influence of sample ageing. However, some key information of the samples history is missing: it has been recently shown that WS2 oxidation is initiated by visible light for samples at ambient conditions. Illumination is unavoidable at both AFM systems used (Bruker Icon and NT-MDT Smena), so we may expect the samples were oxidized. This would be true even for a sample presented for reference in Fig. S1 and labeled as fresh sample. This is not a deficiency of the presented research, only the sample history should be described better in this regard.
Likewise the humidity during the measurements is known to play role (in fact, sensitive humidity sensors have been proposed based using this material).
The results of a broad set of techniques are intriguing and may serve as a reference for future observations of phenomena on this amazing physical system. A reader can only wish that the results would be even more interesting if they could be recorded on the same sample at the same location, ideally also at the same location at different states (fresh, aged, …). This is of course highly challenging and goes beyond the scope of the presented research.
The paper is well written, with no apparent errors. Yet it should be checked once again for minor issues, for example at line 303 the last word should be thicker, not thicken.
Author Response
The stated goal of demonstrating a comprehensive analysis was achieved for samples of WS2 on epi Graphene (epiGr) on SiC substrate, using an impressive set of techniques: two AFMs enabling PF-QNM, UFM, KPFM, DFM plus SEM, microRaman and PL.
- The sample preparation is well described, including the influence of sample ageing. However, some key information of the samples history is missing: it has been recently shown that WS2 oxidation is initiated by visible light for samples at ambient conditions. Illumination is unavoidable at both AFM systems used (Bruker Icon and NT-MDT Smena), so we may expect the samples were oxidized. This would be true even for a sample presented for reference in Fig. S1 and labeled as fresh sample. This is not a deficiency of the presented research, only the sample history should be described better in this regard.
This is absolutely correct. However, this manuscript intends to report on the wealth of information that one can collect using a multi-technique approach. It is therefore concentrated on the analyses of samples that have undergone the same treatment conditions; i.e. exposed for two days to an ambient environment, including light exposure. Prior to any analysis, the samples have been kept under an nitrogen atmosphere and in opaque containers to prevent any light exposure.
In our previous work (see Ref. 6), an accurate study of the degradation process has been reported for a WS2 layer directly grown on epitaxial graphene. In particular, we have kept the sample in a opaque container but in air at around 30% humidity for different months until a complete degradation of the WS2 Layer. The sample has been exposed to direct light only during the SPM analyses. In order to address the reviewer’s comment, we have added a paragraph in the Method section regarding the sample storage (line 69): “The samples were kept in opaque containers under a nitrogen atmosphere prior to any analysis, and in opaque containers in air during the SPM analysis.”
- Likewise the humidity during the measurements is known to play role (in fact, sensitive humidity sensors have been proposed based using this material).
This is also correct, as discussed in our previous paper (see Ref. 6). In addition to that, we can say that we have also performed another experiment keeping a fresh sample in an ambient environment but at a temperature of 80°C, in order to discern the roles of humidity and light. We have observed that, on a time scale of a few hours, the oxidation process seems not to take place. As, at this temperature, water should not be present on the surface, this suggests a more relevant role of humidity in the oxidative process. However, the data collected are not sufficiently conclusive to be published.
- The results of a broad set of techniques are intriguing and may serve as a reference for future observations of phenomena on this amazing physical system. A reader can only wish that the results would be even more interesting if they could be recorded on the same sample at the same location, ideally also at the same location at different states (fresh, aged, …). This is of course highly challenging and goes beyond the scope of the presented research.
We believe that, as far as the sample is stable to air and light exposure, this analysis can be carried out on the same sample provided with markers.
- The paper is well written, with no apparent errors. Yet it should be checked once again for minor issues, for example at line 303 the last word should be thicker, not thicken.
We have amended the text at line 303 and read again the whole manuscript. Some of the changes made are highlighted in yellow.
Round 2
Reviewer 1 Report
I read the authors’ response and checked the changes introduced in the manuscript. I think the paper is now ready for publication. However, I still do not understand the differences in mechanical properties between the trenches and the WS2 regions.